# Bridging the Semantic Gap Between Text and Table: A Case Study on NL2SQL

**Lin Long**[♡*], **Xijun Gu**[♡*], **Xinjie Sun**[♡], **Wentao Ye**[♡], **Haobo Wang**[♡◇], **Sai Wu**[♡◇],
**Gang Chen**[♡], **Junbo Zhao**[♡†]
[♡]Zhejiang University, [◇]Hangzhou High-Tech Zone (Binjiang) Institute of Blockchain and Data Security

## Abstract

The rise of Large Language Models (LLMs) has revolutionized numerous domains, yet these models still exhibit weakness in understanding structured tabular data. Although the growing context window promises to accommodate a larger volume of table contents, it does not inherently improve the model's ability to understand the underlying structure and semantics of tabular data. To bridge the semantic gap between **T**ext and **T**able, we propose TNT, a table-language model that features multimodal table representations to empower LLMs to effectively and efficiently abstract structure-enriched semantics from tabular data. TNT also introduces a scalable and efficient training pipeline, featuring novel self-supervised tasks, to integrate abstract tabular knowledge into the language modality. Extensive experimental results on NL2SQL demonstrate a much better table understanding of TNT, which achieves up to **14.4%** higher execution accuracy compared with traditional text-based table representations.

## 1 Introduction

The emergence of Large Language Models (LLMs) has reshaped the landscape of deep learning (Brown et al., 2020; Touvron et al., 2023). Researchers have been actively exploring extending LLMs' capabilities to process images, videos, and audio, broadening their applicability across diverse domains (Liu et al., 2023; Hu et al., 2024; Zhan et al., 2024; Chen et al., 2023). Tabular data, given its ubiquity and unique application value, has also attracted increasing interest from the community (Ruan et al., 2024; Lu et al., 2024b; Fang et al., 2024). However, the special properties and structures of tabular data pose distinct challenges for LLMs, raising a fundamental question: *How well can LLMs truly understand tabular data?*

To investigate this, we focus on NL2SQL, a pivotal task in table understanding where LLMs translate natural language queries into SQL, given a specific database (Pourreza & Rafiei, 2023; Gao et al., 2024; Dong et al., 2023). It requires an abstract semantic understanding of relational tables and often serves as an interface between LLMs and tables in mainstream table-related methods (Cheng et al., 2023; Ye et al., 2023; Jiang et al., 2023b; Wang et al., 2024). To harness LLM for tabular tasks, a straightforward yet common strategy is to serialize each column with its name, data type, attributes, and sample cell values, into prompts (Li et al., 2024b; Pourreza & Rafiei, 2023). However, we find that based on such naive representation, ***current LLMs struggle to form a robust and consistent understanding of comprehensive table semantics***. As shown in Figure 1, model performance varies significantly with different cell value selections, rather than generally benefiting from a comprehensive understanding of the given tables (Figure 1). It suggests that the model resorts to flat context mapping rather than real table structure-aware semantic comprehension – an issue also noted in prior work (Li et al., 2024d; Yang et al., 2022).

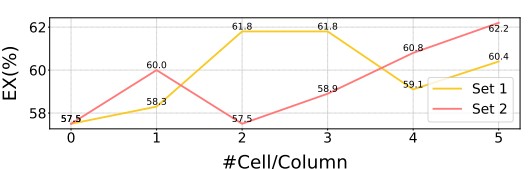

Figure 1: Performance of GPT-3.5 Turbo on Spider-Realisitc with different sets of cell values serialized into the prompt.

*Equal contribution.    [†] Correspondence: `j.zhao@zju.edu.cn`.

Such a phenomenon drives us to rethink the limitations of text-based table serialization (Zhang et al., 2024; Li et al., 2024c), which can be summarized into three key aspects: 1) **Structural Incompatibility**: The bi-dimensional structure of tabular data is inherently misaligned with the sequential, autoregressive nature of language models, making LLMs less sensitive to structure-related patterns within its context; 2) **Redundancy**: Common serialization methods for tabular data are highly token-inefficient, especially when dealing with large tables or databases, leading to huge computational overhead and potential performance degradation due to long contexts; 3) **Bias**: Serializing potentially uncurated table schemas along with a limited number of cell values into the prompt can cause the model to overfit to the local information presented in the context, leading to an incorrect or incomplete understanding of the table's semantics.

To bridge this gap and make LLMs better table readers, we believe that a better table representation is needed — one that can abstract the structure-enriched semantics of original tables. Although earlier works have explored learning table representations with smaller models like BERT and BART (Yin et al., 2020; Liu et al., 2022b), these approaches are less competitive today due to their limited language understanding capabilities. Drawing inspiration from Multimodal Large Language Models (MLLMs) that integrate features from various modalities with text features (Liu et al., 2023; Bai et al., 2023; Lu et al., 2024a), we introduce TNT, a novel framework that empowers LLMs to extract and reason over high-level representations of tabular data, treating tabular inputs as a distinct modality. As depicted in Figure 3, TNT comprises three components: a structure-aware **Table Encoder**, which captures the abstract semantics of tabular contents and converts it into compact embeddings through a bi-dimensional architecture; a **Table-Language Adaptor**, which maps these embeddings into the LLM's textual space using learnable queries (Bai et al., 2023; Tong et al., 2024); and an **LLM** decoder, which performs multimodal table reasoning on downstream tasks.

The proposed tabular representation method introduces another critical challenge: How can we effectively train and align these embeddings-based representations so that they not only encode valuable tabular information but are also interpretable and usable by the LLM? To address this, TNT follows a three-stage training strategy: **Encoder Pre-training**, **Table-language Feature Alignment**, and **Instruction Tuning**. In the encoder pre-training stage, we introduce column-wise contrastive learning, allowing the Table Encoder to learn general column semantics using unlabeled tables. For feature alignment, we create two synthetic table-language interleaved datasets, supplemented by adaptations of existing datasets (Nan et al., 2022; Pasupat & Liang, 2015; Parikh et al., 2020), to teach the LLMs how to leverage column embeddings to address textual instructions. Finally, in the instruction tuning stage, we fine-tune the model using labeled data to enhance its instruction-following capability for specific downstream tasks. For better clarity, we will focus on the NL2SQL task as a concrete example, which can also serve as an interface in other general table-related tasks.

Our main contributions are threefold:

- **(Insights)** We highlight the limitations of LLMs in understanding structured tabular data, introducing novel representations to enhance the understanding on abstract table semantics.

- **(Methodology)** We propose TNT, a multimodal framework, along with an efficient training pipeline, paving the way towards scalable, structure-enriched table semantics learning.

- **(Experiments)** We conduct extensive experiments on NL2SQL under challenging setups that resemble real-world tables, to verify the effectiveness and generalizability of TNT.

## 2 MOTIVATION

**Understanding table content matters.** As discussed above, current LLMs struggle to fully interpret serialized tabule contents, while their comprehension often relies heavily on hand-crafted meta-information (e.g., column names, descriptions, foreign keys) (Sui et al., 2024a). However, we argue that such meta-information is insufficient and unreliable to provide comprehensive table semantics, whereas a structure-aware understanding of tabular contents is essential for developing a complete and accurate grasp of table semantics (Yin et al., 2020). A common challenge in real-world tables is that they usually come with uncurated schemas, where column names may be ambiguous, while detailed descriptions are missing. For instance, abbreviations are frequently used as column names, yet their meanings can vary depending on the context. Some tables may even include non-semantic placeholders as column names (Figure 2). In such cases, understanding table structure and fully

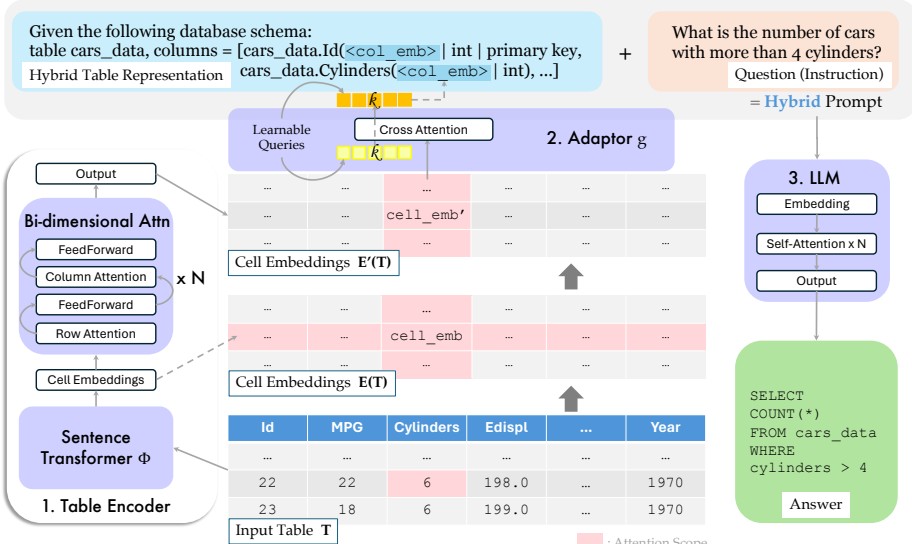

Figure 3: Overall architecture of TNT. The Table Encoder generates cell embeddings from input tables, which are aggregated at the column level by the Adaptor to be further utilized by LLM.

interpreting the patterns and interrelationships within its contents (i.e., *abstract table semantics*) are key to accurate table understanding. This not only reduces the model's reliance on concrete manual annotations but also enhances its robustness and overall table comprehension. However, as shown in Figure 2, LLMs struggle to understand the serialization of tables with non-semantic schemas, which is the gap that TNT aims to fill.

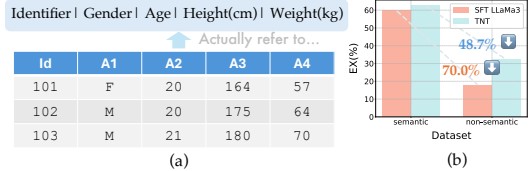

(a)                    (b)

Figure 2: (a) Table with non-semantic column names. (b) Performance on SPIDER-Realistic, where SFT LLAMA3 uses a text-based table representation.

**LLMs need better table representations.** Although prior studies have explored learning table representations (Yin et al., 2020; Liu et al., 2022b; Reimers & Gurevych, 2019), each has limitations to present a generalized form of table-modality knowledge that can be effectively utilized by LLMs. Given the constraints of text-based representations discussed above, we believe that an effective and alignable table representation should ensure: 1) **Structure-awareness.** Considering the bi-dimensional structure of tabular data, a comprehensive and robust table representation should capture semantics across cell, row, column, and table dimensions while being invariant to permutations of rows and columns. 2) **Expressive Efficiency.** To handle tables of varying sizes, an effective table representation should also be capable of appropriately abstracting and compressing table information based on proper semantic segmentation, to ensure expressive efficiency for better utilization by LLMs. 3) **Cross-table Generalizability.** An effective and robust table representation should not only be able to encode table-specific information but also incorporate common tabular knowledge to ensure generalizability across different tables, thus better enhancing LLMs' abstract table understanding. To satisfy these requirements, our paper introduces a dedicated model architecture that progressively derives column-wise semantics, coupled with a specialized training strategy to enable the model to learn effective table representations that can be seamlessly aligned with LLMs.

## 3 ARCHITECTURE

As illustrated in Figure 3, TNT is an end-to-end table-language model consisting of three components: 1) the **Table Encoder**, which generates structure-enriched semantic embeddings based on

the input tables; 2) the **Table-Language Adaptor**, which further aggregates the output embeddings from Table Encoder and aligns them with textual features; and 3) the **Large Language Model (LLM)**, which leverages the high-level table representations and reasons within the bi-modal space.

**Table Encoding.**    Tables consist of heterogeneous cells that are formally independent yet logically organized. To more effectively *abstract* and *encode* this type of semantics, we introduce a Table Encoder, taking the unique structural characteristics of tabular data into consideration. Formally, let $\mathbf{T} = [\mathbf{c}_{11}, \ldots, \mathbf{c}_{1n}; \ldots, \mathbf{c}_{mn}]$ represent a table with $m$ rows and $n$ columns, where $\mathbf{c}_{ij}$ denotes the cell content in the $i$-th row and $j$-th column. The Table Encoder starts with a sentence transformer $\Phi$ to extract semantic representations from each cell, transforming the variable-length cell contents into uniform, compact "cell tokens":

$$\mathbf{E}(\mathbf{T}) = [\Phi(\mathbf{c}_{11}), \ldots, \Phi(\mathbf{c}_{mn})] \in \mathbb{R}^{m \times n \times d}, \tag{1}$$

where $d$ is the dimension of each cell embedding. These cell embeddings are then passed through a stack of bi-dimensional attention modules (Zhu et al., 2023; Somepalli et al., 2021), where they interact with other relevant cells to capture global structural semantics:

$$\mathbf{E}'(\mathbf{T}) = \text{2D-Attn}(\mathbf{E}(\mathbf{T})) \in \mathbb{R}^{m \times n \times d}. \tag{2}$$

Within each module, BERT-style bi-directional attention (Devlin et al., 2019) is applied first along each row and then along each column. This alternating attention mechanism enables the model to capture both the distributional properties within individual columns and the interrelationships between columns. To maintain the permutation invariance of (relational) tables, positional embeddings are intentionally excluded from these bi-dimensional attention modules.

*Remark.*   Similar to the role of a vision encoder in a Vision-Language Model (VLM) (Liu et al., 2022a; Bai et al., 2023; Tong et al., 2024), the Table Encoder leverages tabular knowledge acquired through large-scale pre-training (discussed in later sections) to transform the input table into a series of cell embeddings that encapsulate the structure-enriched global semantics of the table, analogous to patch embeddings in image encoding. These cell embeddings serve as the fundamental informative units, which can be aggregated to form more compact and comprehensive table representations.

**Table-Language Adaption.**   Given the varying sizes of tables, the cell embeddings derived from the Table Encoder can be highly redundant and are not naturally aligned with the textual inputs of LLMs. Therefore, *aggregation* and *alignment* are needed to make a more effective and efficient utilization of these embeddings. To ensure expressive efficiency under appropriate semantic segmentation, we introduce an Adaptor $g$, to aggregate cell embeddings at the column level, as columns are considered the fundamental units that define a table's semantics. Specifically, $g$ performs cross-attention between $k$ learnable queries (Bai et al., 2023; Tong et al., 2024; Li et al., 2023c) and cell embeddings from each column, transforming cell embeddings from tables with an arbitrary number of rows ($m$) into fixed-length ($k$), column-wise compact representations that are aligned with the LLM's embedding dimensionality ($d'$). The embedding of $i$-th column can be derived by:

$$\mathbf{C}(\mathbf{T})_i = g([\mathbf{E}'(\mathbf{T})_{1i}, \ldots, \mathbf{E}'(\mathbf{T})_{mi}]) \in \mathbb{R}^{k \times d'}. \tag{3}$$

Being jointly trained with the LLM on table-language interleaved data, the Adaptor generates representations that can be more effectively processed and interpreted by the specific LLM, thereby enhancing its reasoning and generation capabilities for downstream tasks.

**Dynamic Context Integration.**   In TNT, we differentiate between the abstract semantics that can be represented by column embeddings and other table information involving concrete details (eg., column names and foreign keys), which are inherently sequential and better preserved in text form to ensure precise generation by the LLM. To effectively *integrate* both embedding-based and textual representations, we adapt a widely used prompt template (Li et al., 2024b; Yin et al., 2020; Pourreza & Rafiei, 2023), transforming it into a **hybrid table representation**:

"table `tab_name`, columns=[`tab_name.col_name`(`<col_emb>`|`dtype`|`if_primary_key`)]" .

During inference, the respective column embeddings are dynamically inserted into the specified slots at the LLM's embedding layer. This hybrid representation relieves the Table Encoder from losslessly compressing all table details, while still providing the LLM with valuable high-level insights. It seamlessly combines the abstraction of column embeddings with the specificity of text-based information, resulting in a more comprehensive and expressive table representation.

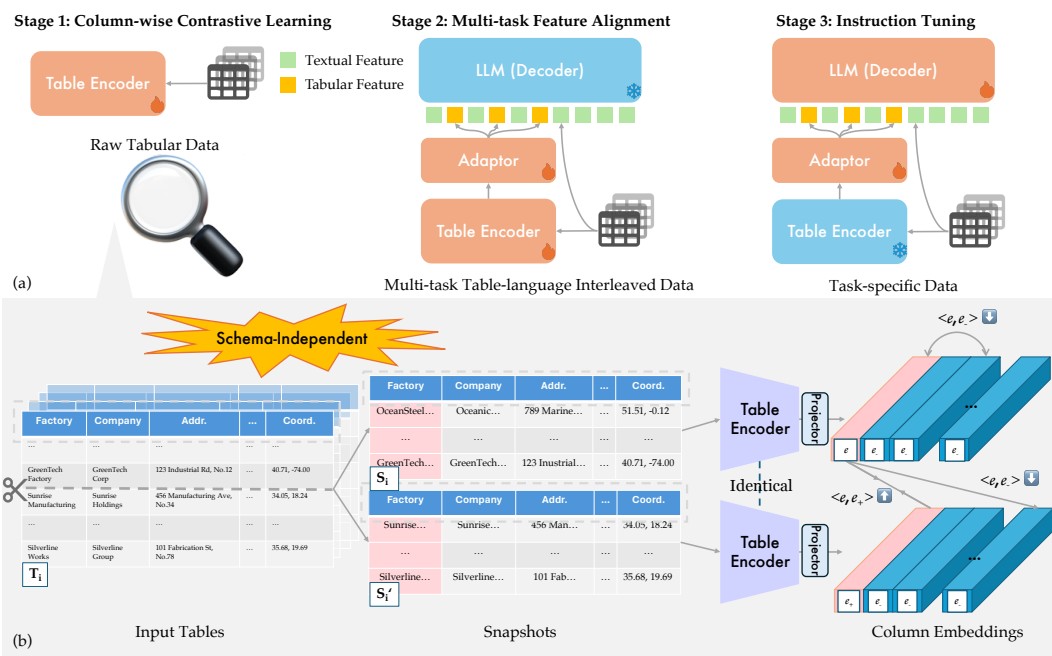

Figure 4: (a) Overall training process of TNT. (b) Detailed depiction of column-wise contrastive learning. $e_+$ and $e_-$ denote positive and negative samples w.r.t. $e$, respectively.

## 4    TRAINING

Similar to other MLLMs, the training logic of TNT falls into three parts: 1) **Encoder Pre-training**, where we introduce a novel *column-wise contrastive learning* objective to independently train the Table Encoder on large-scale raw table data, enabling it to acquire general tabular knowledge; 2) **Feature Alignment**, where we leverage multi-task table-language interleaved data to establish a conceptual linkage between tabular and textual features; and 3) **Instruction Tuning**, where we fine-tune the model on task-specific labeled data, to improve TNT's instruction-following capability for targeted downstream applications.

**Data.** To support large-scale training in Stages 1 and 2, we collect a dataset $\mathcal{D}$ of $86,046$ high-quality business tables from various domains, including finance, education, and medicine, ensuring broad coverage of the complexity and diversity of real-world tables. To facilitate modality fusion, we incorporate table-language interleaved data from existing benchmarks, including FetaQA (Nan et al., 2022), WikiTableQuestion (Pasupat & Liang, 2015), and ToTTo (Parikh et al., 2020).

### 4.1    COLUMN-WISE CONTRASTIVE LEARNING

Humans naturally identify patterns within columns and distinguish differences between columns, which represents the most essential and general form of table semantics. Inspired by this, we employ contrastive learning to explicitly guide the Table Encoder to extract features that unify the intra-column semantics while differentiating between columns, using only unlabeled tables in a schema-independent manner. As shown in Figure 4, we first apply random row sampling on each table $\mathbf{T}_i$ within the mini-batch, creating two snapshots, $\mathbf{S}_i$ and $\mathbf{S}'_i$, which share the same schema but have different cell contents. The Table Encoder then generates an embedding pool $P$ consisting of column embeddings from each snapshot in the mini-batch. During contrastive learning, positive pairs are formed by the embeddings from the same columns across the two snapshots. Following Chen et al. (2020a), we maximize the similarity between positive pairs while minimizing the similarity of other negative pairs using the InfoNCE loss (more details provided in Appendix B.3), which is defined as:

$$\mathcal{L}_{\text{cont}}(\tau, P) = -\frac{1}{|P|} \sum_{e \in P} \log \frac{\exp(e^{\top} e_+ / \tau)}{\sum_{e' \in P \setminus \{e\}} \exp(e^{\top} e' / \tau)}, \tag{4}$$

where $e_+$ is the embedding of the same column as $e$ but from a different snapshot, and $\tau$ is the temperature. This strategy drives the model to learn discriminative and context-aware column embeddings, capturing the unique semantics of each column within a specific schema.

## 4.2 MULTI-TASK FEATURE ALIGNMENT

In the pre-training stage, the Table Encoder acquires general tabular knowledge through self-supervised learning on large-scale raw tables. However, without alignment on task-driven data, these pre-trained representations often lack interpretability, making it challenging to provide valid information for LLMs' reasoning and generation. To bridge the gap between tabular and linguistic elements within the embedding space, we curate a multi-task table-language interleaved dataset and jointly train the Table Encoder and Adaptor in conjunction with a specific backbone LLM.

We extracted a subset of tables from $\mathcal{D}$ to create training samples for two synthetic tasks: ① **Column Prediction** — predicting which column a given cell value belongs to, and ② **Cell Prediction** — identifying which cell value corresponds to a given column. Solving these tasks requires the model to utilize the information captured in the column embeddings to recognize distributional patterns and interrelationships between columns. To promote reasoning based on column embeddings rather than schema-dependent signals, we manually anonymize $50\%$ of the column names, preventing overfitting on schema-specific cues and fostering a more generalized table comprehension.

To further increase data diversity, we also adapt FetaQA (Nan et al., 2022), WikiTableQuestion (Pasupat & Liang, 2015), and ToTTo (Parikh et al., 2020) into formats suitable for training column embeddings, resulting in three adapted tasks: ③ **Question Generation** — generating a question on based on a given answer from a specific table, ④ **Table Titling** — creating a brief title for a given table, and ⑤ **Row Summarization** — summarizing the content of a specific row. To align with our goal of semantic table understanding, we ensure that these outputs maintain an appropriate level of abstraction by minimizing their reliance on detailed cell values. In total, we curate $292,235$ samples for feature alignment. Detailed statistics are provided in Appendix B.2.

## 4.3 INSTRUCTION TUNING

Although we employ diverse prompt templates during the feature alignment stage, some loss of general instruction-following capabilities in the LLM is still inevitable. Therefore, to refine the LLM's comprehension of column embeddings and ensure optimal performance in downstream tasks, we further unfreeze the backbone LLM and conduct supervised fine-tuning using finely labeled data. To optimize training efficiency, only the Adaptor and LLM are trained during this stage.

## 5 A MULTIMODAL PERSPECTIVE

From the introduction above, we can see that TNT integrates several design principles inspired by existing MLLMs, including: 1) **A Structure-optimized Encoder**: Like Vision Transformers with special partitioning and 2D positional embeddings (Caron et al., 2021), TNT utilzes a Table Encoder with a specialized bi-dimensional attention mechanism that directly captures the row-column structures in tabular data. 2) **Hierarchical Feature Extraction**: The cell-to-column feature extraction in TNT mirrors the hierarchical feature process in vision and text models, (e.g., patch-to-image and work-to-sentence) (Bai et al., 2023; Reimers & Gurevych, 2019). This approach combines local details with global features while achieving token efficiency. 3) **Scalable Encoder Pre-training**: Previous works that apply common MLM or contrastive learning to tabular data (Zhu et al., 2023; Wang & Sun, 2022) typically impose high requirements on table curation, thereby limiting their generalizability. The proposed column-wise contrastive learning objective is instead resilient to most table variations and can well generalize to a broader range of tables, showing greater potential in scalability. 4) **Knowledge Injection via Modal Fusion**: Incorporating additional modalities breaks the limitations of homogenized input processing in uni-modal LLMs from both architectural and training perspectives, which cannot be accomplished by language-driven tuning-based methods.

In summary, our work provides a pioneering validation of the feasibility of integrating tabular and textual modalities. We hope to offer a perspective from tabular data for future efforts to unify all data modalities. More discussions are in Appendix E.

Table 1: Comparative results between TNT and variants using traditional text-based table representations on datasets with *non-semantic* column names. **Best performance** on each setup is highlighted.

| Backbone | Method | Dev | | Test | | DK | | Realistic | |
|---|---|---|---|---|---|---|---|---|---|
| | | EM | EX | EM | EX | EM | EX | EM | EX |
| LLAMA3-8B-Instruct (AI@Meta, 2024) | Original | 12.8 | 21.9 | 18.3 | 40.5 | 11.6 | 31.0 | 7.1 | 14.2 |
| | SFT | 30.9 | 31.7 | 43.1 | 50.5 | 21.7 | 30.8 | 19.1 | 17.9 |
| | TNT | **46.0** | **44.6** | **51.4** | **59.9** | **36.3** | **44.7** | **35.6** | **32.3** |
| MISTRAL-7B-Instruct (Jiang et al., 2023a) | Original | 8.5 | 17.9 | 14.0 | 33.9 | 6.2 | 23.2 | 5.1 | 7.9 |
| | SFT | 32.8 | 32.5 | 44.2 | 52.4 | 23.6 | 33.8 | 20.5 | 16.9 |
| | TNT | **43.1** | **42.6** | **51.4** | **57.4** | **34.0** | **41.7** | **36.8** | **32.9** |
| CODELLAMA-7B-Instruct (Rozière et al., 2023) | Original | 9.1 | 22.5 | 17.9 | 39.8 | 8.6 | 29.0 | 6.7 | 12.0 |
| | SFT | 32.1 | 32.6 | 43.1 | 51.7 | 24.3 | 33.1 | 21.5 | 20.9 |
| | TNT | **39.3** | **40.1** | **47.9** | **56.6** | **31.4** | **39.6** | **32.5** | **28.9** |

# 6 EXPERIMENTS

## 6.1 EXPERIMENTAL SETUP

**Datasets.** In this paper, we evaluate TNT on the task of NL2SQL. We adopt three prevalant datasets, **SPIDER** (Yu et al., 2018), **SPIDER-DK** (Gan et al., 2021a), and **SPIDER-Realistic** (Deng et al., 2021). SPIDER is a large-scale cross-domain NL2SQL dataset containing $10,181$ annotated questions that correspond to $5,693$ distinct SQL queries and 200 multi-table databases with *semantic* column names. It is divided into three folds: training, development, and test sets. We reserve data (including tables) in development and test sets for evaluation, ensuring they are unseen during training (Xie et al., 2024; Qu et al., 2024). SPIDER-DK and SPIDER-Realistic are more challenging variants of the original SPIDER dataset. SPIDER-DK emphasizes the importance of domain knowledge, while SPIDER-Realistic removes explicit mentions of column names in the questions.

Ideally, robust table understanding should not overly depend on the quality of column names (i.e., whether they are meaningful or not). To assess the models' abilities to understand table contents independently of column names, we create a ***non-semantic* version** of each dataset, in which $80\%$ of the column names are anonymized. This modification forces the models to leverage structural and contextual clues within table contents, rather than relying on direct meanings of column names, to accurately infer the meaning of each column and thereby generate correct answers.

**Metrics.** Consistent with prior work (Xie et al., 2024; Qu et al., 2024), we evaluate our framework using two metrics: Exact Set Match Accuracy (EM) and Execution Accuracy (EX). EM measures the exact match between keywords in the predicted SQL query and the gold query, while EX compares the execution results of the predicted SQL query with that of the ground truth SQL query on some database instances, which provides a more precise estimate of the model's performance. We use the official test suite from Zhong et al. (2020) for evaluation.

**Implementation Details.** The Table Encoder integrates a sentence transformer initialized with all-MiniLM-L6-v2 (Wang et al., 2020), a lightweight BERT-based text encoder. We set the number of learnable queries $k$ in the Adaptor to $5$, which yields the most robust performance empirically. We evaluate TNT using three mainstream open-source LLMs: LLAMA3-8B-Instruct (AI@Meta, 2024)[1], MISTRAL-7B-Instruct (Jiang et al., 2023a)[2], and CODELLAMA-7B-Instruct (Rozière et al., 2023)[3] as backbone models (results are based on LLAMA3-8B-Instruct unless otherwise specified). To ensure a fair comparison, we set the decoding temperature to $0$ to eliminate randomness during evaluation. Instruction tuning is conducted on the training set of SPIDER. More detailed setups are provided in Appendix B. The code is available at: `https://github.com/llong-cs/tnt`.

---

[1]`https://huggingface.co/meta-llama/Meta-Llama-3-8B-Instruct`
[2]`https://huggingface.co/mistralai/Mistral-7B-Instruct-v0.3`
[3]`https://huggingface.co/codellama/CodeLlama-7b-Instruct-hf`

Table 2: Comparative results on standard NL2SQL benchmarks with *semantic* column names.

| Backbone | Method | Dev | | Test | | DK | | Realistic | |
|---|---|---|---|---|---|---|---|---|---|
| | | EM | EX | EM | EX | EM | EX | EM | EX |
| LLAMA3-8B-Instruct (AI@Meta, 2024) | Original | 34.0 | 54.1 | 32.5 | 58.0 | 29.5 | 50.8 | 25.6 | 54.7 |
| | SFT | 73.0 | 71.5 | 71.9 | 77.8 | 57.4 | 63.7 | 63.4 | 59.6 |
| | TNT | **73.2** | **72.7** | **72.6** | **79.0** | **58.5** | **65.4** | **66.1** | **62.6** |
| MISTRAL-7B-Instruct (Jiang et al., 2023a) | Original | 26.5 | 46.7 | 26.2 | 53.7 | 21.5 | 46.7 | 21.9 | 35.6 |
| | SFT | 69.7 | 68.2 | 68.6 | 75.6 | 50.7 | 58.7 | **64.2** | 57.3 |
| | TNT | **72.4** | **71.3** | **70.1** | **78.6** | **57.9** | **63.6** | 64.0 | **59.1** |
| CODELLAMA-7B-Instruct (Rozière et al., 2023) | Original | 29.0 | 51.2 | 31.0 | 59.7 | 24.9 | 51.0 | 26.6 | 38.6 |
| | SFT | 65.7 | 65.0 | 64.7 | 72.6 | 53.3 | 58.9 | 53.3 | 54.5 |
| | TNT | **66.2** | **66.2** | **65.3** | **73.5** | **54.3** | **60.6** | **56.7** | **55.5** |

## 6.2 OVERALL PERFORMANCE

**Effectiveness on Structure-aware Semantic Understanding of Table Contents.** While humans can effortlessly interpret a table even with ambiguous or irrelevant column names, traditional LLMs often struggle under these conditions. To demonstrate the proposed multimodal representation provides LLM with insights on abstract table semantics, we evaluate its performance against the original and fine-tuned versions of backbone LLMs on datasets with *non-semantic* column names, providing no sample cell values but column embeddings. As shown in Table 1, TNT dominantly outperforms all the baselines on both metrics, achieving up to **16.5%** higher EM and **14.4%** higher EX. These results indicate that TNT can derive insights from the structure-enriched semantics in column embeddings, accurately recognizing the distributional patterns within each column and their interrelationships, thereby inferring the reference of each column for precise SQL generation. This demonstrates TNT's strengths as a robust table interpreter, going beyond mere schema parsing.

**Generalizability across Various Setups.** We further assess the generalizability of TNT using standard academic NL2SQL benchmarks. As shown in Table 2, TNT consistently surpasses the baselines across most benchmarks for all backbone LLMs, demonstrating strong generalizability. These results indicate that even when table schemas are semantically informative, a deeper structure-enriched semantic understanding of table contents provides additional insights. This underscores the value of extracting abstract semantic features from table contents using high-level column embeddings, which are not effectively represented through text-based approaches. Additionally, we observe slight variations in performance gains between instruction-tuned LLMs and code-tuned LLMs, which may be attributed to differences in their generalizability, thereby affecting the effectiveness of feature alignment. We leave further optimization of the backbone models for future work.

## 6.3 ABLATION STUDIES

TNT involves a multi-stage training pipeline, therefore, we conduct ablation experiments to evaluate the significance of each stage. Specifically, we remove each training step independently from TNT and assess the resulting impact on the model performance. As shown in Table 3, removing any of the three training stages leads to a notable performance decline (up to **-23.4%**), underscoring the unique and essential contributions of each stage: 1) **Encoder Pre-training (PT)** equips the model with a foundational tabular knowledge through cross-table self-supervised learning. Without this explicit guidance for distinguishing column semantics, the embeddings risk overfitting to spurious task-specific patterns and less diverse tables, resulting in less generalizable and informative embeddings. As shown in Figure 5, without PT, TNT shows a sharper decline in the loss during feature alignment under the same learning rate, suggesting that it may be exploiting shortcuts instead of building a robust understanding of table semantics. 2) **Multi-task Feature Alignment (FA)** bridges the gap between tabular and textual features with table-language interleaved data. Skipping FA and proceeding directly to instruction tuning significantly weakens the robust integration between Table Encoder and LLM, due to the lack of data diversity. 3) The final stage, **Instruction Tuning (IT)**, is

Table 3: Ablation results on training process.

| Col Emb | PT | FA | IT | Dev | Test |
|:---:|:---:|:---:|:---:|:---:|:---:|
| ✓ | | ✓ | ✓ | 36.0 | 55.8 |
| ✓ | ✓ | ✗ | ✓ | 31.9 | 53.4 |
| ✓ | ✓ | ✓ | | 21.2 | 39.2 |
| ✗ | | | ✓ | 31.7 | 50.5 |
| ✓ | ✓ | ✓ | ✓ | **44.6** | **59.9** |

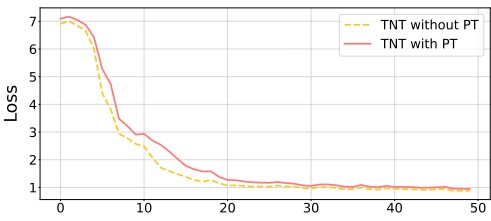

Figure 5: Loss curve during feature alignment.

critical for enhancing the model's instruction-following ability, without which the model is not able to generate informed responses even when the earlier stages provide strong column embeddings.

## 6.4 FURTHER ANALYSIS

**Column embeddings are not soft prompts.** Due to the introduction of additional learnable parameters, one may question whether the column embeddings function merely as soft prompts rather than being truly tied to table semantics. To address this concern, we compare TNT with LLMs fine-tuned with soft prompts, using the same number of learnable parameters as TNT during instruction tunings. As shown in Table 4, the introduction of soft prompts do not bring significant performance boost, likely because they do not contribute to understanding table semantics but instead focus on enhancing the model's task-specific abilities, which are already largely addressed by instruction tuning. We also

Table 4: EX Accuracy with different forms of learnable representations on *non-semantic* SPIDER.

| Method | Dev | Test |
|:---|:---:|:---:|
| SFT LLAMA3-8B-Instruct | 31.7 | 50.5 |
| SFT (w/ Soft Prompts) | 30.4 | 51.5 |
| TransTab (Wang & Sun, 2022) | 23.7 | 33.0 |
| CM2 (Ye et al., 2024) | 17.6 | 39.2 |
| TNT | **44.6** | **59.9** |

evaluate the effect of other embedding-based table representations, such as TransTab (Wang & Sun, 2022) and CM2 (Ye et al., 2024), while both approaches show poor alignment with the backbone LLM. Results above highlight that the column embeddings introduced in TNT not only genuinely capture high-level abstractions of table contents, but also are well alignable in conjunction with LLMs for effective reasoning and comprehension.

Table 5: EX Accuracy on SPIDER-Dev with different numbers (#) of example values.

| Dataset | Method | #=0 | #=1 | #=3 | #=5 | #=20 |
|:---|:---|:---:|:---:|:---:|:---:|:---:|
| Semantic | Original | 54.1 | 57.0 | 57.6 | 57.0 | 57.0 |
| | SFT | 71.5 | 73.1 | 73.5 | 72.3 | 71.8 |
| | TNT (#=1) | | | – **74.0** – | | |
| Non-semantic | Original | 21.9 | 37.7 | 39.9 | 38.4 | 37.8 |
| | SFT | 31.7 | 46.8 | 47.5 | 46.9 | 46.0 |
| | TNT (#=1) | | | – **51.5** – | | |

**Column embeddings tell more than serialized contents.** We make further comparisons between our hybrid representations and traditional text-only representations. Specifically, we analyze the impact of serializing different numbers of sample cell values into the prompt. To minimize potential side effects from changing the prompt template, we include a minimal number of example values (#=1) in TNT. Three key observations emerge from Table 5: 1) Incorporating tabular content generally improves model performance, indicating the significance of table contents. 2) However, model performance does not consistently correlate with the number of cell values included, as LLMs with limitations in structural understanding, struggle to fully comprehend large volumes of table contents. Worse still, incorporating cell values result in a longer context length, which can negatively impact model performance, making the optimization on text-based table representations even

harder. 3) For TNT, using just one example value achieves the best performance across all variants, highlighting that compact column embeddings, which are structure-enriched and more semantically interpretable, provide insights *beyond* serialized table contents. This demonstrates the effectiveness of column embeddings in enhancing the model's ability to process and reason over table contents in a more *token-efficient* manner, overcoming the limitations of text-based approaches.

**Compatibility Analysis.** Regarding the potential of TNT, we demonstrate that the proposed column embeddings are highly compatible with common prompting techniques used in existing NL2SQL methods. As shown in Table 6, we integrate techniques such as Schema Filtering (SF) (Li et al., 2024b), Code Correction (CC) (Sun et al., 2023), and Self-Consistency (SC) (Pourreza & Rafiei, 2023) into TNT, all of which led to noticeable improvements. We can even approach methods that are based on closed-source LLMs like GPT-4 (Gao et al., 2024; Sui et al., 2024b) (86.6%) and GPT-3.5 (Dong et al., 2023) (82.3%), with a much smaller backbone LLM, such as LLAMA3-8B. This, to some extent, also suggests that the structure-enriched semantics captured by column embeddings provide unique value that cannot be effectively substituted by text-based prompting techniques alone.

Table 6: Effects of combining TNT with other techniques on SPIDER-Test.

| Method | | EX |
|---|---|---|
| TNT (LLAMA3-8B-based) | + SF | 79.6 |
| | + CC | 84.3 |
| | + SC | 80.9 |

## 7  RELATED WORK

**Tabular Prompt Design.** Harnessing LLMs for tabular tasks first requires converting tabular data into a format compatible with text-based inputs. A common method is to serialize tables into formats like Markdown, JSON, or XML (Fang et al., 2024; Singha et al., 2023; Sui et al., 2024a). However, prior study reveals that even minor changes, such as row permutations, can lead to substantial performance fluctuations (Liu et al., 2024), indicating that LLMs lack a consistent understanding of table structures. Additionally, due to context length limitations, feeding all table contents into the LLM is often impractical. Some methods retain only schema information (Pourreza & Rafiei, 2023; Gao et al., 2024), which may hinder generalizability. Retrieval or compression methods selectively include critical table content based on predefined rules, which mitigate context constraints (Herzig et al., 2020; Li et al., 2024b). Nevertheless, these strategies, like other prompt-based methods, still fail to overcome the fundamental limitations of LLMs in comprehending structured data.

**Tabular Training.** Table tuning focuses on improving (open-source) LLMs' understanding of tables with a substantial volume of table-related data. For example, Table-GPT (Li et al., 2024c) employs a synthesis-followed-by-augmentation strategy to create datasets for table tuning, while TableLlama (Zhang et al., 2024) uses real-world datasets for instruction tuning on LLAMA2. While effective, these methods still face architectural limitations. Pre LLM-era methods have explored learning table representations with adapted architectures with special positional embeddings (Majmundar et al., 2022; Yin et al., 2020), attention mechanisms (Zhu et al., 2023; Somepalli et al., 2021) or decoding strategies (Liu et al., 2022b), and special learning objectives, including masking (Yin et al., 2020; Herzig et al., 2020), corrupting (Huang et al., 2020) and contrastive learning (Wang & Sun, 2022; Ye et al., 2024). However, these methods typically lack enough generalizability and the reasoning and generation prowess of modern LLMs, which reduces their applicability.

## 8  CONCLUSION

In this paper, we present TNT, a multimodal framework with novel table representations that enhance LLMs' ability to comprehend structure-enriched semantics from tabular data. TNT features a tailored architecture optimized for tabular structures, along with training strategies specifically designed for effective table understanding. Extensive experiments on the NL2SQL task demonstrate significant performance gains, achieving up to **14.4%** higher execution accuracy, underscoring its effectiveness in semantic understanding of tabular data. Our work paves the way toward an effective modality fusion between tabular and textual data. We hope it provides valuable insights for future research.

REPRODUCIBILITY STATEMENT

The implementation details of TNT are outlined in Section 6.1 and further elaborated in Appendix B. Due to copyright and privacy restrictions, some proprietary table data used in our training process cannot be publicly released. However, users can train the model within the same framework using their own table data. A derivative table expert model is available at `https://github.com/tablegpt/tablegpt-agent`.

ACKNOWLEDGEMENT

This paper is supported by NSFC under Grants (No. U24A201401). This work is also partially supported by the Pioneer R&D Program of Zhejiang (No. 2024C01035) and the Fundamental Research Funds for the Central Universities (No. 226-2024-00049).

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

Table 7: Detailed statistics of model architecture and computational overhead per inference.

| # Learnable | Table Encoder | Adaptor | LLM (LLAMA3-8B-Instruct) |
|---|---|---|---|
| Parameters | 102,910,464 | 68,734,976 | 8,030,261,248 |

| Computational | Table Encoding | Feature Alignment | Decoding |
|---|---|---|---|
| Overhead (ms) | 36.49977 | 2.37205 | 693.60648 |

## A   LIMITATIONS AND FUTURE WORK

Due to computational constraints, our experiments are conducted on open-source LLMs with approximately 7B parameters. We anticipate further enhancing TNT by scaling up the training data and evaluating it on more comprehensive benchmarks (e.g., NL2Code) or more table structures (e.g., hierarchical tables) to fully explore its potential. We also hope our work can bring insights to other research focused on enhancing LLMs' ability to understand other forms of structured data.

## B   IMPLEMENTATIONS DETAILS

### B.1   ARCHITECTURE DETAILS

We provide detailed statistics on the number of learnable parameters in each component of TNT and their computational overhead during inference in Table 7. The results show that the additional learnable parameters and computational overhead introduced by TNT are minimal, indicating that TNT maintains high efficiency while enhancing table understanding.

### B.2   TRAINING SETUPS

When it comes to the training details, during the column-wise contrastive learning stage, we pretrain the Table Encoder on the dataset of $86,046$ tables for 20 epochs, using a batch size of $64$ and a constant learning rate of $10^{-4}$. For multi-task feature alignment, the Table Encoder is connected to a downstream LLM through the Adaptor. In this stage, we freeze the LLM and only train the Table Encoder and Adaptor on multi-task data for a single epoch with a batch size of $1024$. A cosine learning rate scheduler is used, with a maximum learning rate of $5 \times 10^{-5}$, a minimum learning rate of $10^{-6}$, and a warmup ratio of $0.05$. Lastly, the instruction tuning for all methods is performed on the training set of SPIDER for 2 epochs with a batch size of 128. We use a cosine learning rate scheduler with a maximum learning rate of $3 \times 10^{-6}$, no minimum learning rate, and a warmup ratio of $0.05$. Additionally, a weight decay coefficient of $0.1$ is applied. For all training stages, we use the AdamW optimizer with $\beta_1 = 0.9$, $\beta_2 = 0.999$, and $\epsilon = 10^{-8}$. The input embeddings of the decoder are always truncated to a maximum length of $4096$. Baseline implementations follow the setups described in previous works (Li et al., 2024b; Gao et al., 2024). All models are trained on 8 A100 GPUs with 80GB of memory. Note that we did not specifically optimize the training hyperparameters; thus, the above configuration may not yield the absolute optimal model performance. We provide detailed statistics about the training data used in the Feature Alignment stage in Table 8.

### B.3   DETAILS OF COLUMN-WISE CONTRASTIVE LEARNING

In this section, we provide a detailed explanation of the calculation of the contrastive loss in Equation 4. During column-wise contrastive learning, we first apply random row sampling on each table $\mathbf{T}_i$ in the mini-batch to generate two snapshots, $\mathbf{S}_i$ and $\mathbf{S}'_i$, which maintain the same schema but differ in cell content. Corresponding columns in the two snapshots are denoted as $\mathbf{c}_i$ and $\mathbf{c}'_i$. The Table Encoder then processes each snapshot to produce column embeddings, which are further normalized using a projection head. This process yields an embedding pool $P$ consisting of column embeddings from each snapshot, across every tables in the mini-batch. Next, we construct positive pairs by matching column embeddings from the same column but different snapshots (i.e., $\mathbf{c}_i$ and $\mathbf{c}'_i$), while embeddings from other columns (which can belong to any tables in the minibatch) in $P$ form negative pairs for a given embedding $\boldsymbol{e}$. We use $\boldsymbol{e}_+$ to denote column embedding that can

Table 8: Details of multi-task pre-training data.

| Task | Data Sources | # | Description | Example Prompt Template |
|---|---|---|---|---|
| Column Prediction | $\mathcal{D}$ (Synthetic) | 50,000 | Predict which column is the cell value in the given row from. | `Given the following database schema:`
`<database_prompt>`
`Suppose there is a row: [<cell_value>,`
`<cell_value>, <cell_value>, ...]`
`Which column is the cell value`
`<cell_value>most likely from? Answer`
`with the corresponding column name only.` |
| Cell Prediction | $\mathcal{D}$ (Synthetic) | 50,000 | Predict which cell value is from the given column. | `Given the following database schema:`
`<database_prompt>`
`Which cell value in [<cell_value>,`
`<cell_value>, ...] is most likely from`
`column <col_name>?`
`Answer with the corresponding cell value`
`name only.` |
| Question Generation | FetaQA, WikiTableQuestion | 20,935 | Ask a question based on the given answer. | `Given the following database schema:`
`<database_prompt>`
`Please propose a relevant question which`
`can be answered with <answer>.` |
| Table Titling | ToTTo | 85,613 | Provide a brief introduction of the given table. | `Given the following database schema:`
`<database_prompt>`
`Suppose the table above is from a web`
`page, please write an introductory title`
`for this table.` |
| Row Summarization | ToTTo | 85,687 | Provide a brief summarization of the given row. | `Given the following database schema:`
`<database_prompt>`
`Please summarize the following rows in one`
`sentence (the columns may be shuffled).`
`<row>` |
| Total | - | 292,235 | - | - |

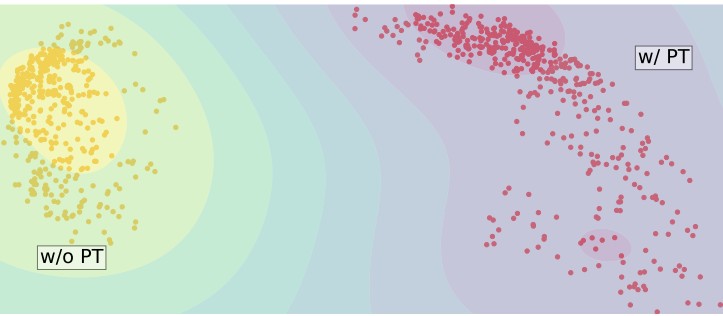

Figure 6: Distributions of column embeddings on SPIDER-Dev, produced by TNT with pre-training and TNT without pre-training.

form a positive pair with $e$. Then, we use Equation 4 to maximize the cosine similarity between positive pairs and minimize it between negative pairs. To further enhance the model's learning, we implement a hard negative sampling strategy within each batch. This strategy prioritizes selecting columns from the same table or database, which are superficially similar and thus harder to distinguish. By focusing on these challenging negatives, the model is encouraged to extract nuanced features and capture table-specific global characteristics, thereby improving its robustness and understanding of column semantics. In this way, we observe that **the effectiveness of column-wise contrastive learning and feature alignment heavily depends on the quality of the tables**, particularly their diversity and the average number of columns per table.

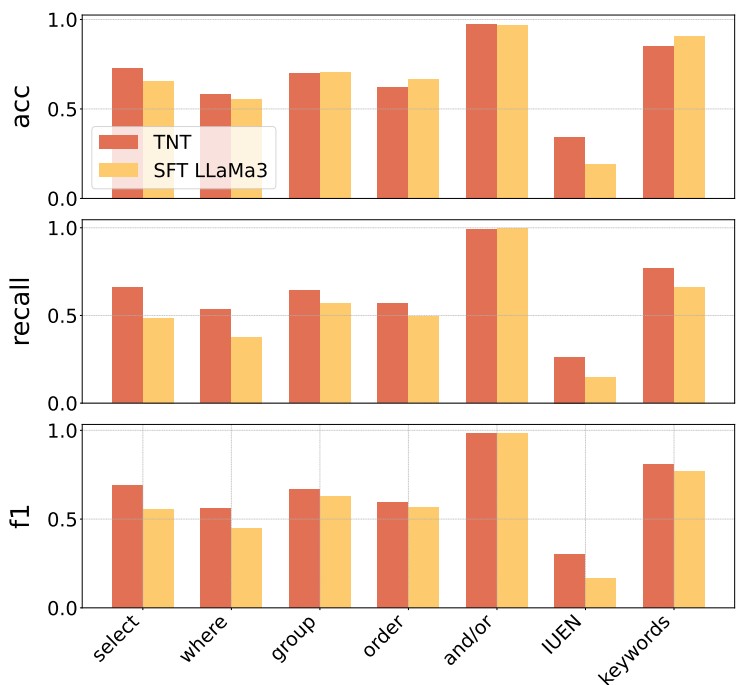

Figure 7: Detailed EM Accuracy of TNT and SFT LLAMA3-Instruct on *non-semantic* SPIDER-Dev.

## C  ADDTIONAL EXPERIMENTS

### C.1  EMBEDDING VISUALIZATION

We visualize the distribution of output column embeddings generated by TNT and TNT without pre-training. As shown in Figure 6, embeddings of different columns produced by TNT without pre-training appear to be more clustered and less discriminative, indicating that, in the absence of pre-training, the column embeddings are prone to overfitting homogeneous task-specific patterns during Feature Alignment and Instruction Tuning. This results in less informative and less valuable embeddings, reducing their effectiveness in enhancing the LLM's table understanding.

### C.2  CORRESPONDENCE BETWEEN COLUMN EMBEDDINGS AND COLUMN SEMANTICS

We also compare TNT with a variant that uses random column embeddings, where the embeddings in each slot have a 30% probability of not corresponding to the actual column names. As shown in Table 9, the effectiveness of TNT is significantly reduced when using mismatched column embeddings, as these embeddings fail to capture valid column semantics and instead mislead the model's reasoning. This demonstrates that the semantics of each column embedding are closely linked to the actual contents in the corresponding columns, which explains why column embeddings help the LLM reason more effectively over table semantics.

Table 9: EX Accuracy with different forms of learnable representations.

| Method | Dev | Test |
|---|---|---|
| SFT | 31.7 | 50.5 |
| SFT (w/ Soft Prompts) | 30.4 | 51.5 |
| TNT (w/ Random Col Embs) | 34.6 | 54.4 |
| TransTab (Wang & Sun, 2022) | 23.7 | 33.0 |
| CM2 (Ye et al., 2024) | 17.6 | 39.2 |
| TNT | **44.6** | **59.9** |

Table 10: Performance of TNT BIRD-dev. The reported metric is execution accuracy.

| Method | BIRD-dev (w/o knowledge ) | BIRD-dev (w/ knowledge) |
|---|---|---|
| Original (QWEN2.5-7B) | 18.97 | 31.42 |
| SFT (QWEN2.5-7B) | 28.49 | 44.07 |
| TNT (QWEN2.5-7B) | **31.42** | **49.28** |
| ChatGPT | 24.05 | 37.22 |

Table 11: Performance of TNT adapted to an agent workflow on WikiTQ and TabFact. The reported metrics in both benchmarks are accuracy (%).

| Method | WikiTQ | TabFact |
|---|---|---|
| TNT (QWEN2.5-7B) | **61.42** | 77.80 |
| Text-to-SQL (GPT-3.5) | 52.90 | 64.71 |
| Binder (GPT-3.5) | 56.74 | 79.17 |
| Dater (GPT-3.5) | 52.81 | 78.01 |
| Chain-of-Table (GPT-3.5) | 59.94 | **80.20** |

## C.3 ADDITIONAL BENCHMARKS

To better demonstrate the generalizability of our method, we follow the same model architecture and training recipe introduced in previous sections (but with different training data) to extend TNT to other general tabular tasks. Specifically, we first evaluate TNT and corresponding variants on BIRD-dev (Li et al., 2023b). BIRD, unlike Spider, emphasizes advanced SQL syntax, the use of external knowledge, and SQL efficiency. Achieving SoTA performance typically requires strong code generation capabilities – often gained through large-scale pre-training on code data – or complex pipeline designs (Li et al., 2024a; Maamari et al., 2024; Pourreza et al., 2024; Li et al., 2024b), which are beyond the tabular representation focus of our work. Despite that, from the results in Table 10 we can still observe a significant improvement by leveraging better table representations with TNT, which demonstrates its generalizability on challenging tasks. Additionally, following a code/SQL-driven agent workflow similar to the ones introduced in Wu et al. (2024); Rajkumar et al. (2022), we further extend TNT to a broader set of tabular tasks, including TableQA, fact verification, numerical reasoning, and data analysis. We evaluate it on TableBench (Wu et al., 2024), HybridQA (Chen et al., 2020c), FEVEROUS (Aly et al., 2021), TabFact (Chen et al., 2020b), and WikiTQ (Pasupat & Liang, 2015), compared with additional baselines (Wu et al., 2024; Rajkumar et al., 2022; Cheng et al., 2023; Ye et al., 2023; Wang et al., 2024). Experimental results on WikiTQ and TabFact (some of the numbers are reported from Wang et al. (2024)) in Table 11 show that TNT achieves performance on par with SoTA methods without any task-specific prompting tricks or complex pipeline designs. Notably, TNT is based on a 7B open-source LLM, whereas other baselines are built on GPT-3.5. Evaluation on other benchmarks shows a similar result in Table 12.

Table 12: Performance of TNT adapted to an agent workflow on TableBench, HybridQA, and FEVEROUS.

| Method | TableBench (PoT@1) | HybridQA (Acc) | FEVEROUS (Acc) |
|---|---|---|---|
| TableLLM (LLAMA3.1-8B) | 25.80 | 27.61 | 42.30 |
| TableLLM (LLAMA3-8B) | 31.96 | 27.35 | 51.50 |
| TableLLM (QWEN2-7B) | 21.05 | 27.12 | 20.10 |
| TableLLM (CODEQWEN-7B) | 26.39 | 20.14 | 46.90 |
| TableLLM (DEEPSEEK-7B) | 28.39 | 19.53 | 18.39 |
| QWEN2.5-Instruct-7B | 22.58 | 51.13 | 63.30 |
| QWEN2.5-Coder-7B | 16.15 | 51.10 | 60.70 |
| TNT (QWEN2.5-7B) | **36.64** | **53.17** | **78.05** |

Table 13: Performance of TNT variant compared with SoTA methods on the Spider leaderboard.

| Method | Spider-test (EX) |
|---|---|
| TNT + LLAMA3-8B + Code-Correction | 84.3 |
| MiniSeek (Anonymous) | 91.2 |
| DAIL-SQL + GPT-4 + Self-Consistency (Gao et al., 2024) | 86.6 |
| DAIL-SQL + GPT-4 (Gao et al., 2024) | 86.2 |
| DIN-SQL + GPT-4 (Pourreza & Rafiei, 2023) | 85.3 |
| C3 + ChatGPT + Zero-Shot (Dong et al., 2023) | 82.3 |
| RESDSQL-3B + NatSQL (Li et al., 2023a; Gan et al., 2021b) | 79.9 |

## C.4 ADDITIONAL METHOD COMPARISONS

In this section, we provide additional method comparisons between TNT variants and other SoTA methods on the Spider leaderboard. Results in Table 13 show that TNT shows competitive performance with simple prompting techniques and a relatively small-scale model.

## D CASE STUDY

To more intuitively demonstrate the effect of TNT, we summarized cases where the standard SFT model failed but TNT succeeded. The differences between the two models are primarily reflected in four areas: **schema linking**, **column hallucination**, **content matching**, and **content reasoning**. Representative examples are shown in Figure 8. Notably, the SFT LLAMA3-Instruct model, lacking a strong understanding of database contents, often struggles to align SQL conditional statements with the actual database representations. As illustrated by Figure 7, TNT consistently outperforms the SFT model on critical keywords such as "SELECT" and "WHERE", which are essential for showcasing the model's schema linking capabilities. Besides, the SFT model frequently fails to recognize special representations in the database, such as using "T" to denote "True" in the bottom-right example. It may also hallucinate column names that do not exist in the schema. These phenomena highlight the SFT model's weak grasp of the overall structural understanding of tables.

## E A MULTIMODAL PERSPECTIVE (ELABORATED)

From the above introduction, we can see that TNT integrates several design principles inspired by existing MMLMs, including:

**A Structurally-Designed Encoder.** Similar to how Vision Transformers (ViT) in VLMs utilize image partitioning and 2D positional embeddings (), TNT leverages a Table Encoder with a specialized two-dimensional attention mechanism that directly captures the unique structural properties of tables. This approach better encodes row-column correspondences, leading to more accurate semantic representations of the table's contents.

**Progressive Feature Extraction.** Existing MMLMs typically follow a hierarchical encoding process (e.g., from patch to image, from word to sentence) (), which integrates local details with global features while maintaining token efficiency. TNT follows a similar strategy by first using the Table Encoder to produce cell-level embeddings that capture structure-enriched semantics and then aggregating these embeddings through an Adaptor into compact and token-efficient column-level representations.

**Scalable Encoder Pre-training.** Achieving scalable pre-training requires robust training objectives that handle diverse input quality. Previous works that apply common MLM or contrastive learning to tabular data () typically impose high requirements on table curation, thereby limiting their generalizability. To address this, we propose a novel column-wise contrastive learning objective that imposes no constraints on the semantic clarity of column names or the presence of explicit

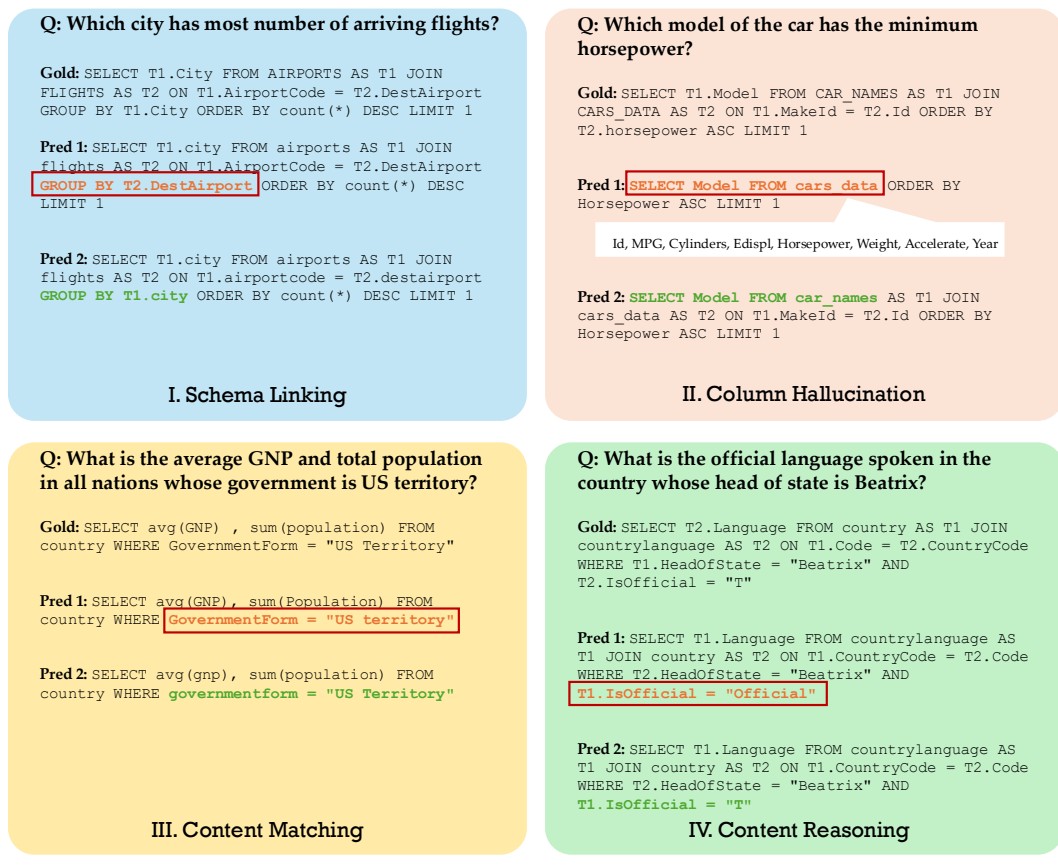

Figure 8: Case study on Dev. Pred 1 is from SFT LLAMA3-Instruct and Pred 2 is from TNT.

inter-column relationships. This makes it more suitable for scaling across larger and more diverse tabular datasets.

**Knowledge Injection via Modal Fusion.** Different modalities of data have distinct characteristics and require different approaches for effective understanding. Introducing additional modalities allows us to integrate the modality-specific comprehension capabilities of custom-designed encoders with the superior language understanding and reasoning abilities of LLMs. This multimodal integration helps overcome the limitations of homogenized input processing in uni-modal LLMs and is fundamentally distinct from traditional tuning-based methods. We argue that this approach is more cost-efficient and offers a higher potential for performance improvements.

# F LIMITATIONS OF EXISTING TABLE REPRESENTATION LEARNING

1) **Structure-awareness.** Some methods directly model serialized tabular data using language models (Liu et al., 2022b), resorting to a linear comprehension that fails to capture the structural information inherent in tables. 2) **Expressive Efficiency.** Some methods model the table content along with other contexts, producing token-level embeddings (Ye et al., 2024). However, these embeddings often lack a coherent hierarchical structure to be compressed effectively, which become fragmented and lack efficiency on larger tables. 3) **Generalizability.** Prevalant tabular pre-training objectives, such as Masked Language Modeling (Zhu et al., 2023) and Column Name Prediction (Yin et al., 2020), rely on associations between cells or between cells and column names. However, they struggle to generalize to tables with more heterogeneous and noisy elements. Lacking any of the above characteristics undermines the effectiveness of table representations, while TNT achieves a balance with a dedicated model architecture and tailored training strategy. To address these issues, TNT introduces a dedicated model architecture that progressively derives column-wise semantics, coupled

with a specialized training strategy to enable the model to learn effective table representations that can be seamlessly aligned with LLMs.

