# OpenReview forum: "Bridging the Semantic Gap Between Text and Table: A Case Study on NL2SQL"
_ICLR.cc/2025/Conference — ICLR 2025 Poster_

### Official Review · Reviewer_HfF2 · 2024-10-21

**Soundness:** 3
**Presentation:** 3
**Contribution:** 2
**Rating:** 8
**Confidence:** 4

**Summary:**

The paper discusses the limitations of Large Language Models (LLMs) in processing structured tabular data, highlighting that broader context windows in next-generation LLMs do not necessarily enhance their capability to comprehend the structural and semantic aspects of tables. To address this gap, the authors introduce TNT, a table-language model designed to integrate the structural information of tables into LLMs. TNT features a robust and scalable training framework, employing contrastive learning to merge tabular data knowledge with linguistic modalities efficiently. The efficacy of TNT is especially showcased through its performance on the NL2SQL task, where it significantly outperforms traditional text-based table serialization methods, achieving up to 14.4% higher execution accuracy.

**Strengths:**

* The idea of treating tabular inputs as a distinct modality is interesting, however, it may somehow be overlapped with conventional text features encoding since the majority of tables are plain text, and the only difference is that tabular data has structure which may inherently be unaligned with the current autoregressive nature of LLMs.
* The motivation for addressing non-semantic column names is sound, although I have concerns about whether these ambiguity issues can be easily resolved by adding schemas or by retrieving relevant documents as supplementary information in LLMs' inference stage.
* The paper is well-written and easy to follow.

**Weaknesses:**

W1: I suggest you rephrase the InfoNCE loss mentioned in Section 4.1 and Figure 4 in case not all readers are familiar with contrastive learning.

W2: The performance of the proposed TNT seems to be limited, especially when considering the leaderboard of Spider (https://yale-lily.github.io/spider). It seems like when equipping GPT-3.5 with chain-of-thoughts or self-consistency can already achieve quite similar or even better performance than the proposed TNT. If so, the major benefits of TNT is that it can support the non-semantic column names; however, maybe this can be easily solved by adding schemas or by retrieving relevant documents as supplementary information. And I also have concerns whether this can be easily solved with more advanced models using inference like GPT-4 or O1 models as well.

Several relevant papers should be included in the references:

* Chain-of-Table: Evolving Tables in the Reasoning Chain for Table Understanding (https://arxiv.org/abs/2401.04398)
* TAP4LLM: Table Provider on Sampling, Augmenting, and Packing Semi-structured Data for Large Language Model Reasoning (https://arxiv.org/abs/2312.09039)
* StructGPT: A General Framework for Large Language Models to Reason over Structured Data (https://arxiv.org/abs/2305.09645)
* Large Language Models are Versatile Decomposers: Decompose Evidence and Questions for Table-based Reasoning (https://arxiv.org/abs/2301.13808)

**Questions:**

Q1: What is the metric used in Figure 1 (what does Ex(%) refer to?) and what do set1 and set2 refer to?

Q2: For the contrastive learning, it's not very clear how the positive and negative pairs are collected. I'm thinking whether you mean that (1) you first sample two snapshots $S_i$ and $S_i^{'}$ using random row sampling, then for each snapshot, you calculate the embeddings for each column, for example denoted as $c_j$ and $c_j^{'}$, and then you consider the positive pair as ($c_j$, $c_j^{'}$) if they share the same column, and consider the pair of $c_j$ with other column embeddings as negative pairs?

Q3: Perhaps a question of rigor. I'm curious about whether the proposed method can be applied to some more challenging tasks like HiTab, which are hierarchical tables. It seems the proposed column-based contrastive learning cannot be simply applied to hierarchical tables.

Q4: What are the datasets used for instruction-tuning mentioned in Section 4.3?

Q5: What are the distributions of the datasets you used for pre-training? It seems the portion of anonymized column prediction samples is quite relevant to why TNT performs better on non-semantic versions of datasets.

Q6: What does "original" refer to in Table 2? Does it mean few-shot inference or some other prompting methodologies? This question is critical since I refer to the leaderboard of Spider (https://yale-lily.github.io/spider). It seems like when equipping GPT-3.5 with chain-of-thoughts or self-consistency, it can already achieve quite similar or even better performance than the proposed TNT. If so, the major benefits of TNT is that it can support non-semantic column names; however, maybe this can be easily solved by adding schemas or retrieving relevant documents as supplementary information. And I also have concerns whether this can be easily solved with more advanced models using inference like GPT-4 or O1 models as well.

Q7: For Table 3, it's puzzling why the performance trends on the development set do not match those on the test set. For example, when only instruction tuning (IT) is used, omitting column embeddings, the scores are 31.7 on the dev set versus 50.5 on the test set. However, when both feature alignment (FA) and IT are applied, the performance is nearly the same on the dev set at 31.9 but improves to 53.4 on the test set.

Q8: For the hybrid table representation mentioned in line 210, is there any ablation study? Is "dtype" an essential property for LM understanding, especially for the NL2SQL task?

---

### Official Review · Reviewer_WAuU · 2024-10-24

**Soundness:** 3
**Presentation:** 3
**Contribution:** 2
**Rating:** 5
**Confidence:** 5

**Summary:**

This paper proposes a so-called TNT model, which comprises three major components, a table encoder similar to the image encoder that encodes the table; a table-language adaptor, which maps the representations from the table encoder to LLM's textual space; and the standard LLM decoder.
To train these components, the paper conducts encoder pre-training, table-language feature alignment, and instruction tuning.
The authors conduct experiments on three text-to-SQL datasets, including Spider, Spider-DK, and Spider-Realistic (the latter two are variants built upon the standard Spider dataset).
The authors have improved the performance compared to the standard supervised fine-tuning on the three datasets and conducted ablation studies.

**Strengths:**

- There seems to be lots of training efforts done by the authors, as they need to pre-train the encoder, align it with the decoder and instruction-tune the model.
- There is some improvement compared to the standard supervised fine-tuning on these datasets.
- There are ablation studies conducted to help understand the contributions of each component.

**Weaknesses:**

- The authors only conduct their experiments on three datasets, all focused on text-to-SQL, which extremely limits the scope of this paper, which undermines their claim in the title as `bridging the semantic gap between text and table`. To be more specific, the authors could have conducted their experiments on more datasets from table QA, table-to-text datasets, etc. Even in the domain of text-to-SQL, they do not conduct any experiments on BIRD benchmark.
- The performance improvement is not significant compared to the SFT baseline. For instance, in Table-2, Llama-3-8B-Instruct achieves 73.0 on Dev while TNT achieves 73.2. This holds for test and other models as well, the improvements are around 0.2, 0.7, 0.6, etc., which makes me wonder whether the improvement indeed is significant.
- The overall performance on Spider is low compared to the state-of-the-art method. On the Spider leaderboard: https://yale-lily.github.io/spider, the best performed method is 91.2 on test set. Even consider the fact that the authors are using the 8B model, the RESDSQL-3B + NatSQL [1], which is a 3B method, achieves 79.9 on the test set, higher than the number reported in the paper (79.0). In addition, RESDSQL-3B + NatSQL came out in 2023.

## References
[1] Li, Haoyang, et al. "Resdsql: Decoupling schema linking and skeleton parsing for text-to-sql." Proceedings of the AAAI Conference on Artificial Intelligence. Vol. 37. No. 11. 2023.

**Questions:**

- Why do you limit your studies to text-to-SQL datasets, especially the Spider dataset and its variants? How does TNT perform on other table-related tasks?
- Why do you claim `bridging the semantic gap between text and table` in your title? Do you consider it as an over-claim considering the scope of your experiments?
- Have you checked the performance listed on Spider leaderboard and include those papers in the related works section?

---

> ### Comment · Reviewer_WAuU · 2024-11-21
>
> I appreciate the effort made by the authors through the detailed response.
>
> The authors claim that *broader table-related task (e.g., Table QA and Table-to-Text) would introduce confounding factors, including pipeline design, numerical reasoning capabilities, and reliance on external knowledge*. However, as claimed by the authors, they aim to study **table representation**, which those downstream table tasks are clearly highly relevant. To some extent, these table tasks are even more relevant than the text-to-SQL dataset experimented by the authors in their main body of the paper, as text-to-SQL involves not only the table, but also the SQL programming ability of the LLMs. If the authors want to argue about the **confounding factor**, have you studied the **confounding factor** for the SQL programming ability of the LLMs? In your paper, you have experimented with Llama 3 8B Instruct and CodeLlama 7B Instruct, which clearly is not an apple to apple comparison as CodeLlama 7B Instruct is based on Llama 2 7B model. I recommend the authors to consider carefully of the points they want to raise in their paper before conducting the experiments.
>
> I appreciate the authors showing more experimental results. However, things become more confusing as the authors are showing the results based on QWen 2.5 7B model, which I did not see them using in the original version of the paper on BIRD dataset. What is your motivation of suddenly turning to QWen 2.5 rather than the ones you used before?
>
> In terms of your results on TableBench, TabFact, WikiTQ, I appreciate you showing more results on table specific tasks. However, I **disagree** with the authors about their claim that *TNT achieves performance on par with SoTA methods **without any task-specific optimizations or prompting tricks***. In your paper (line 251-255), you mention that you incorporate data from existing benchmarks, including FeTaQA, WikiTQ, ToTTo. It seems that WikiTQ is among one of the datasets you used for training. I suggest the authors to be more rigorous about the experimental setups, in both the experiment process and articulating what you have carried out.
>
> In terms of the presentation of the results for Spider, if you do not think EM is a good metric, you may report it in the appendix rather than placing them in the main body of the paper -- the common assumption for results in the main body is that they are important and convey meaningful information, which seems to contradict to your purpose here. Also, speaking of the exact match scores, since you have fine-tuned the model on the Spider dataset, the limited performance improvement for EM scores is a bit confusing to me. Is it because the model does not capture the domain patterns in its fine-tuning process? It would be better if you can provide more explanations as you have put the results in the main body of the paper.
>
> In terms of the variants for Spider dataset -- Please note that they are mainly proposed for the purpose of diagnosing model's weakness. With that said, the scope they cover, and the pattern diversity cannot compare to the commonly adopted ones such as Spider and BIRD. The authors may consider moving those results into some analysis or ablation tables rather than presenting them together with Spider in the main result table.
>
> Thanks for the performance comparison table. It would be better if you can include more method comparisons in your paper (I believe the SoTA on Spider is around 90, which is higher than the 86.6 you presented here).
>
> I understand that you are trying to propose a new *perspective on efficient table representations*. However, the current experiment results fall short of comprehensively analyzing the advantages of such a method:
>
> - How efficient is it? It seems that you have put lots of effort in different training stages, and why do you call such a time-consuming and computationally heavy process efficient?
>
> - How does such method compare to directly encoding tables? If people are familiar with terms from multimodality, there are the distinctions between early and late fusion. Your method seems to lie on the late fusion side, while the common practice is passing both table and text together, where the fusion happens naturally by the attention mechanism. Why is your method necessarily more efficient and effective than fusing the table and question together?

---

> > ### Comment · Reviewer_WAuU · 2024-11-26
> >
> > I thank the authors for their response and engagement in the discussion period.
> > However, since my concerns are not solved, I would maintain my score for now.

---

### Official Review · Reviewer_rghh · 2024-10-27

**Soundness:** 3
**Presentation:** 3
**Contribution:** 3
**Rating:** 5
**Confidence:** 4

**Summary:**

The paper proposes TNT, a multi-modal inspired, efficient training pipeline for structure-enriched table semantics learning, given the current limitations of LLMs in understanding structured data. The proposed method shows improved performance on NL2SQL tasks.

**Strengths:**

1. The proposed technique is built upon three pressing challenges of tabular data understanding: the structural information, the varying sizes, and generalize across tables. The proposed TnT method consists of modules that target corresponding modules, such as table encoder and abstract table representation.

2. The training objectives, especially the “multi-task feature alignment” covers potential usage for multiple common table-related tasks, which presumably enhances task generality in downstream.

**Weaknesses:**

1. Lack of baseline comparison: There have been some efforts in training a general table-oriented LM, such as TableGPT and TableLLaMa. Particularly, it would be informative to compare TableLLaMa as a baseline, to the LLaMa version of TNT method proposed. Currently the baselines are purly out-of-the-box model checkpoints and two training variants (SFT and TNT), so it is kind of obvious that further training would bring increases. Adding additional baselines (such as TableLlama) to prove the effectiveness of the proposed training  recipe would be informative.

2. Lack of table tasks: the only evaluated task is NL2SQL. However, to prove that the proposed model is general across tabular-related tasks, it is highly recommended to evaluate a series of most common tabular tasks, such as table fact verification, table-based analysis generation, and table-based question answering. Particularly because the proposed method TNT claims to apply to bi-directional table structures, it would be helpful to actually test on some benchmarks that features bi-directional tables, such as HiTab and AIT-QA.

**Questions:**

1. What is the unique importance of the “column-wise contrastive learning” task? Previous pre-training have tackled objectives at varied granular levels, such as table-level, row-level, sub-table level, cell level; what make the column-wise objective stand out in this work?

---

### Official Review · Reviewer_Y53T · 2024-11-04

**Soundness:** 3
**Presentation:** 3
**Contribution:** 3
**Rating:** 6
**Confidence:** 3

**Summary:**

The paper proposes a new module for LLMs to better understand the tabular data. The new module, TNT, is inspired by the multi-modal LLMs and encodes the tabular information through a BERT-size encoder. The encoded dense features will be transformed by an adapter and then be inserted during the decoding stage of LLMs.

When building the whole pipeline, TNT first pre-trains the encoder and then trains the encoder and the LLM together in an alignment task. Finally, it conducts instruction-tuning for the specific tasks.

The performance on NL2SQL improves remarkably with the newly proposed method.

**Strengths:**

1. Table understanding is important to LLMs. And it is vital to find an efficient and effective way to enable LLMs to understand the information represented in the tabular structure.
2. The proposed method can potentially solve really huge tables regardless of the context size.
3. The whole training pipeline is well explained and is inspiring to related work.

**Weaknesses:**

The main issue with this paper is the limited evaluation tasks and baselines. The paper only considers NL2SQL while ignoring many other table understanding tasks, such as Table QA (e.g. TabFact, WikiTQ). It is true that it's challenging to solve table-based problems with LLMs. It is still necessary to compare TNT with other methods where only LLMs are used, such as agent-based methods [1,2,3].

The comparison is also potentially unfair. TNT is only compared with SFT while TNT requires much more training than SFT. I assume that SFT models are only trained on the training set of the target benchmark, while TNT needs more training during the encoder training and alignment stages.

[1] Cheng, Zhoujun, et al. "Binding language models in symbolic languages." arXiv preprint arXiv:2210.02875 (2022).
[2] Ye, Yunhu, et al. "Large language models are versatile decomposers: Decompose evidence and questions for table-based reasoning." arXiv preprint arXiv:2301.13808 (2023).
[3] Wang, Zilong, et al. "Chain-of-table: Evolving tables in the reasoning chain for table understanding." arXiv preprint arXiv:2401.04398 (2024).

**Questions:**

I am curious to see the performance of TNT over tables of different sizes compared with baselines. This will better support the claim that a longer context may not perfectly solve the table understanding by LLMs.

---

### Meta-Review · Area_Chair_FkDw · 2024-12-17

**Metareview:**

The paper proposes a multi-modal training pipeline for structure-enriched table semantics learning, given the current limitations of LLMs in understanding structured data. The proposed method shows improved performance on NL2SQL tasks. The proposed technique is built upon three pressing challenges of tabular data understanding: the structural information, the varying sizes, and generalize across tables. The proposed TnT method consists of modules that target corresponding modules, such as table encoder and abstract table representation. The proposed method can potentially solve really huge tables regardless of the context size. There are ablation studies conducted to help understand the contributions of each component.

On the downside, the reviewers mainly worried that the paper itself, although was positioned as table reading, actually only focuses on NL2SQL instead of table reading. The authors have accordingly changed the title to reflect the correct (but more narrow) scope of the study. The authors have also added more results in the rebuttal phase to include more tabular tasks in response to the related concerns raised by the reviewers (regarding limited coverage of initially evaluated tasks).

**Additional Comments On Reviewer Discussion:**

Please make sure to incorporate the additional results. Importantly, please make sure that the title is changed to focus on NL2SQL, since the original title was indeed a bit of overclaim.

---

### Decision · Program_Chairs · 2025-01-22

Accept (Poster)